# Studies of NO_2_ Gas-Sensing Characteristics of a Novel Room-Temperature Surface-Photovoltage Gas Sensor Device

**DOI:** 10.3390/s20020408

**Published:** 2020-01-11

**Authors:** Monika Kwoka, Jacek Szuber

**Affiliations:** Silesian University of Technology, Faculty of Automatic Control, Electronics and Computer Science, Department of Cybernetics, Nanotechnology and Data Processing, 44-100 Gliwice, Poland; Jacek.Szuber@polsl.pl

**Keywords:** surface photovoltage gas sensor, room temperature working conditions, gas sensor characteristics, NO_2_ atmosphere

## Abstract

In this work the characteristics of a novel type of room temperature NO_2_ gas sensor device based on the surface photovoltage effect are described. It was shown that for our SPV gas sensor device, using porous sputtered ZnO nanostructured thin films as the active gas sensing electrode material, the basic gas sensor characteristics in a toxic NO_2_ gas atmosphere are strongly dependent on the target NO_2_ gas flow rate. Moreover, it was also confirmed that our SPV gas sensor device is able to detect the lowest NO_2_ relative concentration at the level of 125 ppb, with respect to the commonly assumed signal-to-noise (S/N) ratio, as for the commercial devices.

## 1. Introduction

After more than six decades of development only a few types of gas sensors devices have been constructed. These are based on a variety of chemical and physical methods including electrochemical, catalytic, optical, acoustical and electrical designs. Reviews of the most common types of gas sensors include, among others, those of Comini et al [1] and by Xiu et al [2]. 

Among the electrical methods, the most common conductometric type gas sensor devices (systems) are based on semiconductor metal oxides (MOX), organic materials, carbon nanotubes and conductometric polymers. However, even after five decades of development, they still exhibit some critical and fundamental limitations, such as high-temperature working conditions and related high power consumption. These sensors do have good sensitivity; depending on the gas, usually at the level of ppm, but this is combined with a rather poor selectivity (a trait that can be slightly improved upon by adding noble catalytic metals); in addition, they possess rather poor dynamic parameters (long response and recovery times) [3,4,5,6,7,8,9,10,11,12]. 

In order to overcome the above-mentioned crucial limitations of conductometric MOX gas sensors, some innovative ideas and related approaches have appeared in the literature in recent years describing improvements of their sensing abilities. 

One of these methods exploits the effect-of-work function (contact potential difference (CPD)) variation, which can be measured by using the Kelvin vibrating capacitor as a transducer [13,14]. However, because of rather poor gas sensitivity due to a rather low signal-to-noise (S/N) ratio, this method has received only limited application in studies of the porous semiconductor materials as recently reviewed by Korotcenkov [15]. 

This type of sensor can be improved by using the additional external illumination of the gas sensor material to play the role of measuring electrode, a phenomenon commonly known the surface photovoltage (SPV) effect [16]. However, until now the SPV effect has received surprisingly limited application for the detection of specific gases at the surfaces of selected metal oxide materials [17,18].

In our recent paper [19], we have briefly described the fundamentals of SPV effect and its potential application for gas sensing, together with a proof-of-principle of our novel type SPV gas sensor system based on the Kelvin probe approach using porous ZnO nanostructured thin films as an active electrode, and a flat Cu metallic grid-type as the reference electrode. Moreover, the basic analytical abilities of our SPV gas sensor device for the detection of nitrogen dioxide NO_2_ atmosphere in relative concentrations of up to 1 ppm in synthetic air have been verified and interpreted on the basis of information on local surface chemistry and morphology of porous ZnO nanostructured thin films. 

In this paper the results of systematic studies of the usable gas-sensing characteristics of our new SPV gas sensor device are presented, including the determination of the influence of gas flow rate of NO_2_ at constant relative concentration in synthetic air on the variation of SPV amplitude versus time, as well as the respective gas sensor dynamic parameters, such as response and recovery times. The influence of gas flow rate at constant relative concentration on the gas sensor response characteristics has recently been investigated by several groups, among others by Eklöv et al. [20] for the Pd-MOSFET sensors in H_2_ detection, by Lezzi et al. [21] for the RGTO SnO_2_ film-based conductometric sensor for CO detection, by Utriainen et al. [22] in the comparative analysis of a miniaturized ion mobility spectrometer and metal oxide gas sensor for the detection of toxic organic vapors, by Kevin et al. [23] for SnO_2_ film-based conductometric sensor mainly for CO detection, by Gmür et al. [24] for the metal-oxide-based gas sensor microarrays mainly for isopropanol detection, and by Righettoni et al. [25] for portable WO_3_ gas sensors for breath analysis.

Moreover, we have determined the variation in amplitude of the SPV signal with decreasing relative concentrations of NO_2_ in synthetic air well below 1 ppm level at the chosen middle constant gas flow rate, to attain the lowest NO_2_ relative concentration that can be experimentally detected at the commonly assumed signal-to-noise (N/S) ratio of 3.

## 2. Materials and Methods

In our experiment the recently elaborated SPV gas sensor device [19] was used based on the porous ZnO-nanostructured thin films [26,27] as an active electrode, and the flat Cu metallic grid-type reference electrode mounted inside the rectangular SPV gas sensor measuring chamber of dimensions 20 × 40 × 30 mm, which were matched to the dimensions of the Kelvin probe flat type vibrating capacitor system used. The SPV signal response after UV illumination of the ZnO active electrode by the UV5-400-30 type LED diode was measured by the microcontroller data processing and respective acquisition system [19].

For the determination of the lowest NO_2_ relative concentration in dry synthetic air significantly below the ppm level we used a novel gas dilution, mixing and dosing system, that was designed by our group and constructed by the MEDSON Company (Paczkowo, Poland). Its simplified schematic idea is shown in Figure 1.

In general, this system consists of:-two bottles and related stainless steel channels of NO_2_ toxic gas at two different starting concentrations in synthetic air (50 ppm to reach a relative concentration range above 1 ppm for primary measurements, and 1 ppm to reach a relative concentration range below 1 ppm for finest measurements, respectively), combined with a set of respective cut-off valves; -the gas dilution, mixing and dosing parts based on gas mass flow controllers (MFC)–model Brooks SLA5850 (Brooks Instruments, Hatfek, PA, USA) combined with a set of respective cut-off valves;-gas mixing chamber with a set of baffle peers to reach the well defined precise final constant relative concentration of NO_2_ gas mixture in the synthetic air;-“back pressure” unit based on the gas mass flow controllers (MFC)–model Brooks SLA5820 (Brooks Instrument, Hatfek, PA, USA) as the pressure regulators, combined with two exhaust systems equipped with additional toxic gas washing bottles (to avoid any undesired escape of NO_2_ toxic gas mixture to the surrounding atmosphere).

In relation to the above, it should be added that this system is also equipped with a microcontroller processing unit for its control, combined with the data processing and acquisition, working with the respective software (MEDSON FC).

It should be underlined that in our experiments we used a gas handling procedure allowing the repeatable measuring conditions defined by the specific NO_2_ gas flow rate in combination with the specific NO_2_ relative concentration in synthetic air reaching the SPV gas sensor measuring chamber. It consists of two independent steps, i.e. the controllable value of the NO_2_ gas flow rate which was achieved by using the above described gas dilution, mixing and dosing system (MEDSON), whereas the controllable NO_2_ relative concentration was achieved by using additionally the GasAlert Extreme NO_2_ detector (Honeywell BW Technologies, Calgary, AB, Canada). For the evaluation of relative NO_2_ concentration below 1 ppm (its detection threshold) an extrapolation procedure was used. Finally, between the subsequent gas sensing measurements the SPV gas sensor measuring chamber was rinsed with pure synthetic air for 30 min. 

## 3. Results and Discussion

In our research, we have focused on the working conditions of our highly sensitive SPV gas sensor prototype, which can be divided in the following two steps:-determination of the influence of NO_2_ gas flow rate at constant relative concentration on the gas sensor response characteristics, together with basic gas sensor dynamic parameters;-determination of the lowest NO_2_ relative concentration that can be detected by our SPV gas sensor device with respect to the commonly used signal-to-noise (S/N) ratio, one of the most important usable parameters for most commercial gas detection systems.

### 3.1. Influence of NO_2_ Gas Flow Rate at Constant Relative Concentration on Gas Sensor Response 

In general, taking into account the dynamic aspects of gas adsorption/desorption effects at the surface of gas sensor materials, the gas-sensing characteristics should strongly depend on the flow rate of target gas reaching the gas sensor system measuring chamber. This is related to the fact that with an increased flow rate, the amount of NO_2_ adsorbed at the surface of the gas sensor material also increases. This can be a cause of the variation of shape of response curve towards the corresponding higher response time, as well as the shape of recovery curve towards the corresponding higher recovery time directly related to the favorable conditions for the gas desorption. Thus, in our study we have determined the influence of NO_2_ gas flow rate at constant concentration in synthetic air on the above-mentioned gas sensor response characteristics, together with basic gas sensor dynamic parameters like response and recovery time.

Figure 2 shows the time-dependent variation of amplitude SPV signal as a function of NO_2_ gas flow rate in the range of 100 ÷ 20 mL/min, at the constant relative NO_2_ concentration of 20 ppm in the standard dry synthetic air (with respect to the level in the synthetic air).

One can observe from our SPV gas-sensing characteristics that with the lowering of gas flow rate in the range of 100 ÷ 20 mL/min (at the constant NO_2_ relative concentration of 20 ppm in the standard dry synthetic air) the amplitude of SPV signal decreases only by a factor of less than 2, taking into account the average accuracy of determination of variation of SPV signal at the level of about a single mV, which corresponds to the signal-to-noise (S/N) ratio at the level of 3. The obtained data are summarized in the Table 1. 

Apart from the decreased amplitude of SPV signal, an additional effect is visible, i.e., a slight variation of the shape (slope) of respective gas sensor curves, directly related to the variation of gas sensor dynamic parameters like response and recovery time(s), respectively. 

Figure 3 presents the time-dependent variation in the amplitude of SPV signal for the mean NO_2_ gas flow rate of 60 mL/min, at the constant relative NO_2_ concentration of 20 ppm in the standard dry synthetic air, and respective gas sensing characteristics including dynamic parameters. 

As shown in Figure 3, the response and recovery time for the mean NO_2_ gas flow rate of 60 mL/min, at the constant relative concentration of 20 ppm in the standard dry synthetic air, was ~240 s and ~1400 s, respectively. The respective values of the above-mentioned dynamic parameters estimated for the various gas flow rates are also summarized in Table 1. In general, the dynamic gas sensor parameters for our SPV gas sensor system at the above-mentioned gas flow rate range of 100÷20 mL/min, look rather poor. For instance, the response time decreased only by about 50%. A more evident tendency is observed for the recovery time because it decreased by a factor of ~3. As mentioned above, these effects can be correlated, from one side to the various amount of NO_2_ gas adsorbed at the surface of gas sensor materials, and from a second one, to the more favorable conditions for the desorption of various amounts of NO_2_ gas from the surface of gas sensor materials under the synthetic air exposure at gas flow rate 60 mL/min.

However, it should be noticed that these values have been obtained for our SPV gas sensor system at room temperature. Crucially, it directly confirms that the target gas (NO_2_) used in our experiments was only physically adsorbed onto the inner surfaces of the ZnO nanostructured thin films used as the gas sensor material. In such conditions a better dynamic characteristics could not be expected. Nevertheless, these dynamic characteristics are better than those of the commonly used conductometric gas sensors for which the response signal at room-temperature working conditions are usually close to the signal-to-noise ratio. Of course, one can expect that recovery time of our gas sensor system will be lowered (i.e., recovery effects can be faster), but only after application of the additional electrode degassing mechanism (e.g., heat or light-based). However, the main advantage of our SPV gas sensor system would be lost. Moreover, further study is required in order to optimize the degassing conditions of the target gas, which are currently in progress.

### 3.2. Threshold Sensitivity of the SPV Gas Sensor System in NO_2_ Atmosphere

As our second objective we concentrated on determining the lowest NO_2_ relative concentration in synthetic air that can be detected by our SPV gas sensor device, with respect to the commonly used signal-to-noise (S/N) ratio. 

Figure 4 presents the time-dependent variation in the amplitude of SPV signal for the lowered relative concentration of NO_2_ target gas below 1 ppm in the standard dry synthetic air, at the mean NO_2_ gas flow rate of 60 mL/min.

As can be seen from Figure 4, the decreased relative concentration of NO_2_ is accompanied by a decreased relative amplitude in the SPV signal, reaching its lowest value for of NO_2_ at the level of 125 ppb, taking into account the commonly used criteria that the level of signal-to-noise (S/N) ratio should not be lower than 3. Crucially, this value was already reached at room temperature, which looks very promising with respect to the performance of commonly used conductometric type gas sensor systems. For instance, similar experiments have been performed by Procek et al. [28] for the conductometric type gas sensor device using similar ZnO nanostructures and the additional UV radiation of similar power density, but working mainly at a higher temperature (~200 C). However, there is only one experimental point on the gas sensor response to 1 ppm of NO_2_ at room temperature, that can be compared with our results. The obtained response was 304 ± 74 (%), which corresponds to similar signal-to-noise (S/N) ratio ≈ 4, as used in our analysis.

In addition to the above, it should be noted that, from our SPV gas-sensing characteristics shown in Figure 4, one also notices that the shape of the response curves is similar. A more precise analysis of their sensor dynamic characteristics proved that the gas response time(s) are similar at the level of about 1000 s, respectively, which was longer with respect to the higher relative concentration of NO_2_ in the synthetic air. 

Of additional importance, the respective recovery time(s) are similar at the level of about 1000 s, and comparable with the gas response time(s). It corresponds to the respective gas sensor dynamic parameters for the gas response curves at lower gas flow rates of NO_2_ at the constant relative concentration in the synthetic air shown in Figure 2. However, the tendency for gas sensor dynamic parameters to be directly related to the controlled adsorption of gas species at the surface of gas sensor material is described above.

## 4. Conclusions

In this study the working conditions of our novel, high-sensitivity gas sensor device using the surface photovoltage effect, based on the Kelvin probe approach, have been elaborated. 

We have focused on the determination of the influence of gas flow rate of NO_2_ at the constant relative concentration of 20 ppm on the sensor response and recovery times, respectively, as well as on the determination of lowest relative NO_2_ concentration below 1 ppm relative concentration in synthetic air that can be detected with this sensor.

Firstly, it was observed that with the lowering of target gas flow rate in the range of 100÷20 mL/min, the amplitude of SPV signal decreases only by a factor of less than 2, taking into account the average accuracy of determination of variation of the SPV signal at the level of about single mV. Moreover, a slight variation in the shape of respective gas sensor curves was observed directly related to a variation of gas sensor dynamic parameters, and almost twice decreasing of the respective response/recovery time(s) was determined.

Secondly, it has been established that, with the lowering of relative concentration of NO_2_ in the standard dry synthetic air, an evident tendency towards decreasing of the relative SPV signal appears, reaching at room temperature a smallest relative NO_2_ concentration at the level of 125 ppb, with respect to signal-to-noise (S/N) ratio at the level of 3. Moreover, the gas sensor response time(s) are similar to the respective recovery time(s) being at the level of about 1000 s.

This study confirms that our SPV gas sensor device [19] has the important advantages that make it potentially suitable for wide practical application. In an ongoing study we plan to improve the performance of our SPV gas detector system by using other specific MOX low dimensional nanostructures recently elaborated by our group [29,30,31] with extended internal surfaces. We anticipate that, with the more-effective adsorption/desorption effects of target gases, one can expect to attain better responses as well as reduction of the response and recovery times. Moreover, we also plan to elaborate the effective source for the faster removal of specific target gas species from the surface of gas sensor material(s) during the regeneration process in our of SPV gas detector system. 

## Figures and Tables

**Figure 1 sensors-20-00408-f001:**
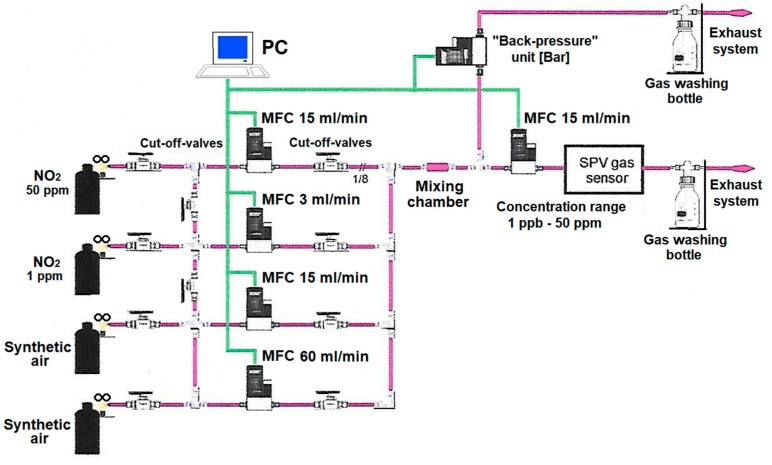
Simplified block-scheme of NO_2_ gas dilution, mixing and dosing system for use with the surface photovoltage (SPV) gas sensor system (the device).

**Figure 2 sensors-20-00408-f002:**
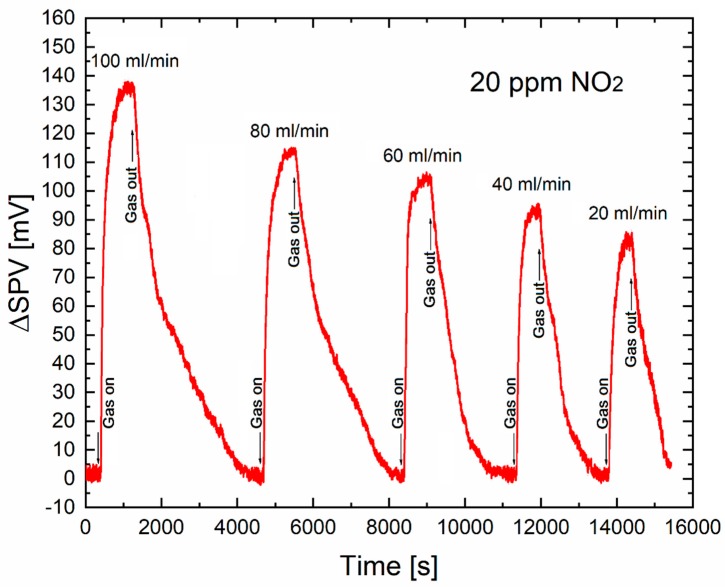
The time-dependent variation of amplitude of SPV signal in NO_2_ at various gas flow rates, and at a constant relative concentration of 20 ppm in synthetic air.

**Figure 3 sensors-20-00408-f003:**
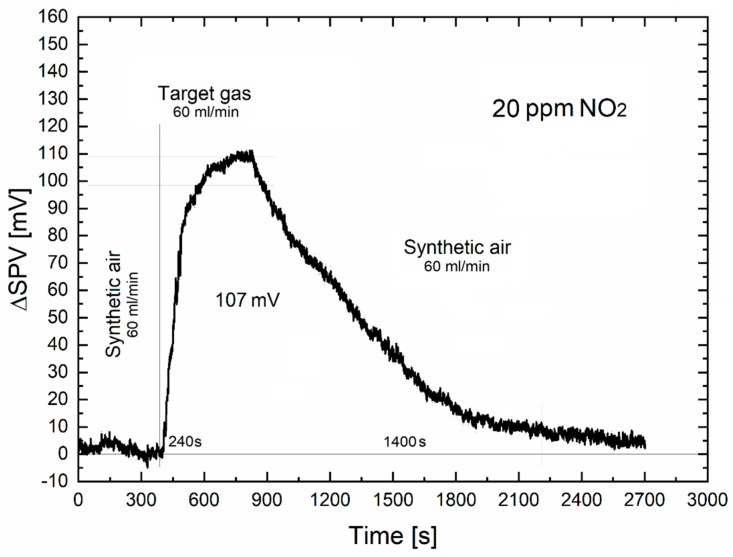
The time-dependent variation of SPV signal for a synthetic air flow rate of 60 mL/min and 20 ppm of NO_2_ concentration.

**Figure 4 sensors-20-00408-f004:**
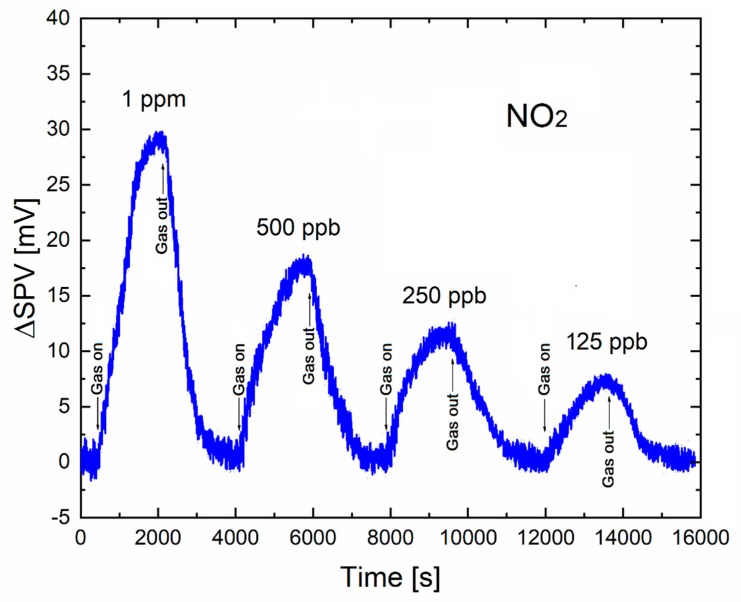
The time-dependent variation in the SPV signal for lowered NO_2_ relative concentration in synthetic air at mean constant gas flow rate 60 mL/min.

**Table 1 sensors-20-00408-t001:** Variation in the SPV signal at various NO_2_ gas flow rates at constant 20 ppm relative concentration, together with the basic dynamic parameters.

Gas Sensor Characteristics		NO_2_ Gas Flow Rate (mL/min)				
		100	80	60	40	20
ΔSPV [mV]		133	119	107	94	81
Dynamic parameters	Response time [s]	~340	~300	~240	~215	~195
	Recovery time [s]	~2400	~2000	~1450	~1200	~900

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
