# Peer review of "Studies of NO2 Gas-Sensing Characteristics of a Novel Room-Temperature Surface-Photovoltage Gas Sensor Device"

_sensors, 2020, doi:10.3390/s20020408_

Round 1
Reviewer 1 Report
In “Studies of Main Gas Sensing Characteristics of Novel 2 Type Room Temperature Surface Photovoltage Gas Sensor Device” authors provided an extension of the work previously published by testing a new mass flow controller apparatus able to reach sub-ppm concentrations for NO2. Thus additional experiments on ZnO nanoporous structures have been conducted investigating the influence of flow rate and the response to NO2 till 125 ppb. In the reviewer opinion some point should be elucidated to better understand the potentialities of proposed manuscript.
Considering the influence of flow rate, the sensor responses increase by increasing the speed of flux; this behaviour is already reported in several works and in some cases it is expected. However, both response time and recovery time of sensor responses increases when the flow rate increases. Reviewer would have expected a decreasing of sensor response time since when the flow rate is higher the time to reach the desired concentration of NO2 in the measurement chamber is faster. For example, in “A facility for characterization and testing of hydrogen sensors” responses are reported in case of different flow rates and the trend is opposite with the one reported here. Please comment this behaviour in the manuscript and, in case, provided the appropriate references. It should be interesting to know the volume of measurement chamber to better understand the influence of flow rate on sensor responses. It should be interesting if authors report a plot showing the sensor responses versus the NO2 concentrations in order to appreciate the linearity of sensor in the interval investigated (125 ppb to 20 ppm). In the manuscript authors state that “Such pretreatment(s) allowed us to remove the undesired adsorbed ambient residual atmospheric gas components including water vapor molecules from the gas sensor material surface playing a role of the gas sensing electrode and enabling a steady reference state for the sensing action.”. Reviewer would like to have a comment on the application of such sensor in a real scenario where humidity is normally present. Do the authors think that the LED irradiation can warm up the ZnO material? In the affirmative case, different flow rate can differently cool down the porous ZnO? Does the temperature affect the sensor baseline or response?
Reviewer 2 Report
The paper is not well written. Major writing revision is required prior to re-submission.
English needs major revision. As a general remark: use simple sentences, for example, lines 33/41 could be split into 4-5 sentences.
Lines 88/90: some important details are missing, e.g., size of magnetron, distance between magnetron and substrate, magnetron power, gas flow rate, deposition time. What kind of target (Zn or ZnO) and gas (presumably Ar or Ar/O2) was employed?
Figure 1: NFC?
Line 119, figure 1, and throughout the paper: “cut-off” valve sounds a bit strange. How about “shut-off” valve? https://www.swagelok.com/en/product/medium-high-pressure-products/medium-high-pressure-needle-shutoff-regulating-valves
Line 179/196 and figure 2: The “Results” part is not understandable at all. For example, nothing “is evident” here nor does it “confirm the general expectations”. These are just “empty” sentences without any meaning. I strongly suggest to (i) to follow the KISS (Keep It Simple and Stupid) rule and (ii) to seek the advice of a native speaker. It is possible to say more and, in particular, more precisely without wasting space for irrelevant statements.
Regarding figure 2 (and figure 4): the flow rate of synthetic air changes but it is not clear at which times. Nor is it explained why the SPV voltage goes up and then down.
Figure 3: “flow rate of target NO2”? In my opinion, “for a synthetic air flow rate of 60 ml/min”. NO2 concentration 20 ppm.
Round 2
Reviewer 1 Report
Reviewer’s comment #1:
Reviewer reports here below a small part of papers that investigated the influence of flow rate on gas sensor response at constant concentration, Please note that in all of these paper response increases with the flux rate.
Utriainen, Mikko, Esko Kärpänoja, and Heikki Paakkanen. "Combining miniaturized ion mobility spectrometer and metal oxide gas sensor for the fast detection of toxic chemical vapors." Sensors and Actuators B: Chemical 93.1-3 (2003): 17-24. Eklöv, Tomas, and Ingemar Lundström. "Gas mixture analysis using a distributed chemical sensor system." Sensors and Actuators B: Chemical 57.1-3 (1999): 274-282. Lezzi, A. M., et al. "Influence of gaseous species transport on the response of solid state gas sensors within enclosures." Sensors and Actuators B: Chemical 78.1-3 (2001): 144-150. Frank, Kevin, Heinz Kohler, and Ulrich Guth. "Influence of the measurement conditions on the sensitivity of SnO2 gas sensors operated thermo-cyclically." Sensors and Actuators B: Chemical 141.2 (2009): 361-369. Gmür, Roman, et al. "Impact of sensor packaging on analytical performance and power consumption of metal oxide based gas sensor microarrays." Sensors and Actuators B: Chemical 127.1 (2007): 107-111. Righettoni, Marco, et al. "Breath acetone monitoring by portable Si: WO3 gas sensors." Analytica chimica acta 738 (2012): 69-75.
In “Lezzi, A. M., et al. "Influence of gaseous species transport on the response of solid state gas sensors within enclosures." Sensors and Actuators B: Chemical 78.1-3 (2001): 144-150”; authors provided a full investigation about the influence of flow rate and chamber geometry on both sensor response and actual concentration in the chamber. Fig. 5 of the aforementioned paper shows that the concentration upon the sensor surface is dependent to the flow rate; just like the time response. This is not surprisingly if we consider that the concentration in the chamber rises faster to the final concentration c0 when the flow rate is higher; then the response time of sensor is supposed to be faster. This behaviour is surely in agreement with the responses reported in “A facility for characterization and testing of hydrogen sensors” paper and supported by data and modelling of the problem.
This is the reason for the reviewer comment and discussion about the behaviour. In the reviewer opinion, a deeper comment and/or a discussion of this behaviour taking in consideration the actual literature on the topic would have benefit the manuscript appeal.
Reviewer’s comment #2:
Authors provided area of chamber (2x3cm) rather than volume. The geometry of chamber and position of inlet and outlet influence the distribution of concentration in the chamber. All of these parameters should be carefully considered, and then reported, in the analysis of dynamic behaviour of sensor responses. (see Lezzi, A. M., et al. "Influence of gaseous species transport on the response of solid state gas sensors within enclosures." Sensors and Actuators B: Chemical 78.1-3 (2001): 144-150” and Gmür, Roman, et al. "Impact of sensor packaging on analytical performance and power consumption of metal oxide based gas sensor microarrays." Sensors and Actuators B: Chemical 127.1 (2007): 107-111.)
Reviewer’s comment #3:
Concerning the concentration vs response plot, since the title of the paper is “Studies of Main Gas Sensing Characteristics of Novel Type Room Temperature Surface Photovoltage Gas Sensor Device”; Reviewer expected an investigation about the main gas sensing characteristics of this class of gas sensor. In the reviewer opinion sensitivity (defined as the first derivative of the curve that fits the sensor responses plotted versus gas concentrations) and linear range should be included if the aim of the manuscript is the one proposed in the title.
Furthermore, if the information about sensor characteristics is already available, then the gas sensor here described cannot be considered novel.
Secondly, the UV activation of gas sensitivity of semiconductor nanostructures is well reported in literature also in the case of conductometric devices. ZnO nanostructures under UV illumination are indeed sensitive to NO2 at room temperature (Procek, Marcin, Agnieszka Stolarczyk, and Tadeusz Pustelny. "Impact of temperature and UV irradiation on dynamics of NO2 sensors based on ZnO nanostructures." Nanomaterials 7.10 (2017): 312.) and in case of heterojunction this material can be very sensitive as proved by Zhou Yong,et al in "UV assisted ultrasensitive trace NO2 gas sensing based on few-layer MoS 2 nanosheet–ZnO nanowire heterojunctions at room temperature." Journal of Materials Chemistry A 6.22 (2018): 10286-10296.” and by Wu Tong, et al. in "UV excitation NO2 gas sensor sensitized by ZnO quantum dots at room temperature." Sensors and Actuators B: Chemical 259 (2018): 526-531. Results presented in literature are far above the noise level (line 220-223).
Despite the kindly responses of authors, Reviewer is still convinced that the main gas sensor characteristics should be deeper investigated and described and that the results obtained should be compared with the results actually present in literature: considering both detection limit and response time for NO2 reported for ZnO nanostructures under UV illumination and references concerning the influence of flow rate for this kind of material and transduction mechanism.
Reviewer 2 Report
The paper is improved and apart from some minor points is now in a publishable form.
General remark: A paper must be self-explaining. Reference to previous papers is definitely helpful but should not be taken as an excuse to omit essential descriptions and details.
Minor points:
Figs 2 and 4: Gas on/off times should be indicated. The gas handling procedure should be briefly described in the text.
line 190: I do not see that the amplitude decrease is proportional to the gas flow. Gas flow reduces by a factor of 5 while amplitude decreases only by a factor of less than 2.
Round 3
Reviewer 1 Report
In the reviewer opinion authors satisfactorily amend the manuscript and comment the reviewer's remarks. The manuscript is worth to be published in the present form.
Author Response
The manuscript has been changed according to the comments